# Zinc(II) Carboxylate Coordination Polymers with Versatile Applications

**DOI:** 10.3390/molecules28031132

**Published:** 2023-01-23

**Authors:** Gina Vasile Scaeteanu, Catalin Maxim, Mihaela Badea, Rodica Olar

**Affiliations:** 1Department of Soil Sciences, University of Agronomical Sciences and Veterinary Medicine, 59 Mărăști Str., 011464 Bucharest, Romania; 2Department of Inorganic, Organic Chemistry, Biochemistry and Catalysis, Faculty of Chemistry, University of Bucharest, 90–92 Panduri Str., 050663 Bucharest, Romania

**Keywords:** zinc, coordination polymer, carboxylate, luminescence, sensor, storage

## Abstract

This review considers the applications of Zn(II) carboxylate-based coordination polymers (Zn-CBCPs), such as sensors, catalysts, species with potential in infections and cancers treatment, as well as storage and drug-carrier materials. The nature of organic luminophores, especially both the rigid carboxylate and the ancillary N-donor bridging ligand, together with the alignment in Zn-CBCPs and their intermolecular interaction modulate the luminescence properties and allow the sensing of a variety of inorganic and organic pollutants. The ability of Zn(II) to act as a good Lewis acid allowed the involvement of Zn-CBCPs either in dye elimination from wastewater through photocatalysis or in pathogenic microorganism or tumor inhibition. In addition, the pores developed inside of the network provided the possibility for some species to store gaseous or liquid molecules, as well as to deliver some drugs for improved treatment.

## 1. Introduction

When we focus on properties, Zn(II) complexes seem unattractive considering both their diamagnetism and lack of color. However, when we evaluate the relationship with living organisms, it can easily be noticed that zinc is the second most abundant transition ion in the series of essential elements, indispensable for a huge number of biological processes. This trace element is critical for cell growth, genomic stability, signal transcription factors, and for the structure and function of a wide range of cellular proteins [1,2,3,4,5].

In thousands of proteins, zinc participates in enzymatic catalysis, structural organization, and/or function regulation. Characteristics such as fast ligand exchange, stereochemical flexibility, and redox inertness sustain its selection in the function of numerous proteins [6]. For these biomolecules, Zn(II) acts mainly as a Lewis acid for enzymes involved in hydrolytic process regulation, as well as a structural factor in both nucleic acid synthesis and their activity control [1,2,3,4,5,6].

Based on the good Lewis acid activity observed in living systems, this ion has found applications in several catalytic processes [7] and in developing species with biological activity [5,8,9]. Moreover, its structural control through coordination generates valuable applications as sensors based on luminescent properties for species with proper selected ligands [10,11,12]. Its stereochemical versatility also allows a structural arrangement in the complex’s network that generates pores and, thus, induces the ability of some materials to storage gaseous, liquid or solid species [13,14,15,16,17,18,19].

Some of these properties are linked to a polymeric structure of complexes that can be achieved by using a proper molar ratio and one or more ligands that can act as a bridge. The ability of monocarboxylate derivatives to link Zn(II) ions into coordination polymers (CPs) by forming two- or three-atom bridges is well established (Figure 1) [20].

Moreover, this ability is enhanced for species containing two or more such groups in their structure. For these species, besides the most common two-atom bridge, there are also bridges in the network structure connecting between three to eight atoms. Hence, several aliphatic and especially aromatic di-, tri- or tetracarboxylate derivatives have been employed for the synthesis of Zn(II)-CPs with diverse structures and various properties. A selection of coordination modes adopted by carboxylate groups in Zn(II) carboxylate-based coordination polymers (CBCPs) is depicted in Figure 1.

Among these materials, metal–organic frameworks (MOFs) represent a special class that provide the advantage of both ultrahigh porosity and specific surface area, high thermal stability, adaptable surface chemistry, adjustable crystal structure and morphology, open metal sites and robustness [18]. The current limitations of Zn-based MOFs in comparison with others Zn-CBCPs come from the fact that is difficult to control the morphology and structure [18] and to create a mixture either of organic linkers, carboxylate and N-donor ligand or of cations Zn(II) and another cation [21].

On the other hand, Zn(II) is suitable for developing such CPs since its d^10^ configuration provides flexible stereochemistry and, consequently, the geometries of complexes can be easily modulated from a tetrahedral geometry through square pyramidal and trigonal bipyramidal geometries to an octahedral geometry, with different distortion degrees, stereochemistry achieved by a proper selection of ligands and molar ratio, respectively.

Furthermore, due to the lability of bonds associated with zero-field stabilization energy in such complexes, the coordinative bond formation is reversible and, thus, enables ligand rearrangement during the polymerization in order to provide diverse ordered networks [20,22]. Consequently, Zn(II) complexes can easily adopt a wide range of 1D, 2D and 3D arrangements, but these also depend on molar ratio, reaction conditions, ancillary ligand nature and coordinative abilities.

A literature survey evidenced several carboxylate derivatives, especially rigid ones such as di- and tricarboxylate benzene derivatives, that are able to develop polymeric species by coordination to Zn(II). Their structure can then be modulated either through self-assembly by carboxylate alone or by using another polydentate species acting as a bridge. These ancillary ligands are usually selected by heterocyclic nitrogen-based species that are also rigid, such as pyridine, imidazole, triazole, bipyridine or phenanthroline derivatives, which, in addition to modulating the topology and porosity, can provide the species with a fluorophore component.

In addition to design interesting structures, the considerable interest in this field is directed towards obtaining CBCPs with useful properties, such as luminescence, porosity, and catalytic, antitumor and antimicrobial abilities, that lead to potential modern applications in several fields, such as sensors, photocatalysts, drugs, or storage materials (Figure 1).

The most recent data concerning these applications are summarized by categories in the following sections.

To fulfill the objectives set for this paper, a literature survey was performed using the most important and commonly accessed databases such as Scopus, Science Direct, and Web of Science. For this approach, keywords according to the paper goal were selected, with zinc coordination polymer being the first used. Afterwards, supplementary keywords such as carboxylate ligand, luminescence, sensor, storage, catalyst, biological activity, and MOF were included in the search.

Based on this procedure, relevant high-quality scientific papers, reviews, and reports were collected, which were then sorted in an organized fashion using the criteria date of publishing, properties and applications (sensors, catalysts, biological activity, storage materials, miscellaneous, etc.) of identified complexes, and carboxylate coordination modes, and, afterwards, subjected to data extraction.

Moreover, some relevant papers identified in accessed databases were used for the backward and forward snowballing technique to identify additional ones which were included in our database and further used for data collection.

Concluding, the data collection resulted from a combination of database search and snowballing. Searches were limited to papers recently published, mainly starting with the year 2012.

## 2. Zinc(II) Carboxylate Coordination Polymers Developed for Different Applications

The compounds reviewed in the following sections show luminescent properties or have robust and thermally stable open-framework structures giving rise to different porosity, which is a valuable characteristic for sorption or selective inclusion of small guest molecules. The photocatalytic abilities were directed to the elimination of drugs or dyes from wastewater, while the biological activity was targeted on pathogenic microorganism or tumor inhibition with the purpose of developing new drugs. Additional applications include the use of Zn(II)-CBCPs as the anode material for lithium-ion batteries (LIBs), in the area of 3D printing or to obtain optoelectronic devices. Further, some species could be used to synthesize nanoparticles with various properties.

### 2.1. Zinc(II) Carboxylate Coordination Polymers with Luminescent Properties

Among water pollutants categories, nitro derivatives, volatile organic compounds, dyes, antibiotics and ionic inorganic species are of major concern because they are noxious for both human health and the environment. Many organic nitro compounds and organic solvents used in industrial activities are of great concern because they could be accidentally discharged into wastewater, leading to environmental unwanted issues. Hence, their detection and sensing require immediate from chemists, this being an aspect that needs to be solved. Among the available methods, fluorescent sensors, characterized by a high sensitivity, fast response, and user-friendly operation, have attracted extensive research interest in recent years [12].

Luminescence of Zn(II)-CBCPs arises from the characteristics of the supplementary organic fluorescent ligands, such as an aromatic conjugated system, while small guest molecules can either enhance or quench this phenomenon. The origin of the luminescence in such species can be related to one or several electron transfer mechanisms, such as intra-ligand (IL), ligand-to-ligand (LLCT), and metal-to-ligand (MLCT) charge transfers [10,11].

If the carboxylate linker displays extended π systems, this will consequently be the source of luminescence. If not, another linker with such characteristics can be engaged in CP synthesis. Moreover, both ligands could have an electron rich π cloud and can, therefore, contribute to luminescence.

Since CPs of Zn(II) have the ability to regulate the emission wavelength of organic materials, several species based on benzene di- or tricarboxylate rigid ligands as tectons were developed as luminescent materials. For some complexes, only the ability to exhibit luminescence was reported, but many of them have proved the ability to act as sensors, either for inorganic or organic species, especially for pollutants originating from industrial and hospital wastes.

Among these, [Zn(1,4-bdc)(H_2_O)_2_]_n_ (**1**) (1,4-H_2_bdc = 1,4-benzenedicarboxylic acid) was synthetized and proved to exhibit blue photoluminescence as a result of intraligand emission [23]. Luminescent properties were also evidenced for 2D-CP [Zn(qnl)_2_]_n_ (**2**) and [Zn_2_(tpp)(OH)_2_(H_2_O)]_n_ (**3**) (Hqnl = quinoline-3-carboxylic acid, H_2_tpp = tetraphenyl phthalic acid) [24], as well as for [Zn(C_4_O_4_)_0.5_(OH)]_n_ (**1a**) which was constructed from squaric acid (H_2_sq) tectons [25].

A systematic study of some factors, namely pH and solvents, led to [Zn_2_(ia)_2_(H_2_O)_4_]_n_ (**4**) and {[Zn_2_(ia)_2_(DMF)]_n_·H_2_O}_n_ (**5**) (H_2_ia = itaconic acid). Interestingly, both compounds exhibit blue fluorescence and green phosphorescence that entails less frequent long-lasting phosphorescence [26].

By introducing different substituents (R = NH_2_, OH, NO_2_) into the1,4-bdc system, structurally diverse CPs [Zn_5_(tib)_2_(bdc-NH_2_)_4_(OH)_2_]_n_·nH_2_O (**6**), [Zn_2_(tib)(bdc-NH_2_)SO_4_(H_2_O)_2_]_n_·5nH_2_O (**7**), [Zn_2_(tib)_2_(Hbdc-OH)_2_(bdc-OH)]_n_·2nH_2_O (**8**), and [Zn(tib)(bdc-NO_2_)]_n_·nH_2_O (**9**) (tib = 1,3,5-tris(1-imidazolyl)benzene) have been synthesized under solvothermal conditions. All species exhibit fluorescence originating from tib [27].

The photochromism was evidenced for [Zn(3-ndi)_0.5_(1,4-ndc)(DMF)]_n_ (**10**) and {[Zn_1.5_(3-ndi)_0.5_(1,4-bdc)_1.5_]·2.5DMF}_n_ (**11**) (3-ndi = N,N’-bis(3-pyridinemethyl)-1,4,5,8-naphthalenediimide, H_2_ndc = 1,4-naphthalenedicarboxylic acid). These complexes also exhibit luminescence assigned to the LMCT, and their fluorescence intensities were gradually reduced upon UV irradiation [28].

Others CBCPs with benzenedi- or tricarboxylate and imidazole [10,29,30,31,32,33], triazole [34,35,36,37], thiazole [38], 2,2′-bipyridine (2,2′-bipy) [39,40], 1,10-phenanthroline (phen) [39,41] or 4,4′-bipyridine (4,4′-bipy) [39,42,43,44,45,46] derivatives were developed for the same purpose.

Furthermore, the same behavior was reported for [(Zn(tactd)_3_(m-tatb)_2_]⋅6H_2_O}_n_ (**12**) (H_3_(m-tatb) = 3,3′,3′′-s-triazine-2,4,6-triyl-tribenzoic acid, tactd = 1,4,8,11-tetraazacyclotetradecane) with luminescence properties arising from the intraligand π–π* transition [47].

All these species are promising luminescent sensors in the detection of various analytes in water and could, as a result, be developed as hybrid inorganic–organic photoactive materials.

However, in addition to the luminescent properties, for other Zn(II)-CBCPs, the sensing abilities for various organic and inorganic water pollutants have been highlighted. These examples, together with corresponding ligands, bridge mode coordination adopted by the carboxylate species, and compound applications, are presented in Table 1.

Hence, a water-stable and luminescent 3D CP {[Zn(ttpa)]⋅1.5DMA}_n_ (**13**) (H_2_ttpa = 2,5-bis-(1,2,4-triazol-1-yl)-terephthalic acid, DMA = N,N-dimethylacetamide) exhibits not only excellent recognition abilities of Fe(III), Cr_2_O_7_^2−^, and antibiotics (nitrofurazone (NFZ), nitrofurantoin (NFT), and dimetridazole (DTZ)) via a luminescence quenching mechanism, but also efficient “turn-on” fluorescence sensing properties of pyridine (py). The detection limits (DLs) are 7.76 × 10^−5^ M for Fe(III); 7.48 μM for Cr_2_O_7_^2−^; and 5.12, 6.57, and 8.59 μM for NFT, NFZ, and DTZ, respectively [48]. The CP [Zn(dttp)(H_2_O)]_n_ (**14**) (H_2_dttp = 2,5-di(1H-1,2,4-triazol-1-yl)terephthalic acid) exhibits an interesting 3D framework combined with high selectivity and sensitivity for Cu(II) over other cations with a high quenching efficiency (Ksv of 4.01 × 10^3^ L·mol^−1^) [49].

For the Zn-CBCP [Zn(piph)(H_2_O)_2_]_n_ (**15**) (H_2_piph = 5-(pyrazin-2-yl)isophthalic acid) synthesized through the hydrothermal method, a 3D framework was evidenced. Its suspension exhibits good luminescence performance in water and, furthermore, the compound shows high selectivity for sensing Fe(III) ion and 2,4,6-trinitrophenol (TNP) in this media. The quenching constants or Stern–Volmer coefficient (Ksv) and the DLs are 1.7 × 10^4^ M^−1^ and 0.44 μM for Fe(III), and 3.8 × 10^4^ M^−1^ and 0.19 μM for TNP, respectively [50].

Interesting structures were evidenced for complexes [Zn(bda)]_n_ (**16**) and [Zn_3_(bta)_2_(H_2_O)_4_]_n_ (**17**) (H_2_bda = 4′-hydroxy-[1,1′-biphenyl]-3,5-dicarboxylic acid, H_3_bta = 4-hydroxy-[1,1′-biphenyl]-3,3′,5,5′-tetracarboxylic acid) synthesized under solvothermal conditions. Both CPs (**16**) and (**17**) exhibited luminescence sensing properties toward trinitrotoluene (TNT) and Fe(III) [51]. Fluorescent nanoscale Zn(II) CP based on 9,10-bis(p-carboxyphenyl) anthracene (bcpa) also exhibits efficient sensing for both nitroaromatic explosive (DNT and TNT) and nitromethane, due to the strong binding affinity to explosive molecules [131].

Another CP, {[Zn_3_(eptc)_1.5_(DMF)_3_]·C_2_H_7_N}_n_ (**18**) (H_4_eptc = 1,1′-ethylbiphenyl-3,3′,5,5′-tetracarboxylic acid), acquired under hydrothermal conditions, displays excellent luminescence and stability in organic solvents. The compound detects triethylamine (TEA) and tetracycline (TET) in water with a fluorescence quenching DL of 1.07 and 0.1 μM, respectively [52].

A highly selectively sensor of Fe(III) was also developed by Tb(III) incorporation into {[Zn(ndic)(H_2_O)_2_]·2H_2_O}_n_ (**19**) (H_2_ndic = 5-(5-norbonene-2,3-dicarboximide)isophthalic acid). Moreover, a paper strip coated with Tb(III)@Zn-MOF material also showed high selectivity for Fe(III) and under UV irradiation [53]. The Fe(III) sensing ability was also evidenced for [ZnLi_2_(cbaiph)(DMF)(H_2_O)]_n_ (**20**) (H_4_cbaiph = 5-(bis(4-carboxybenzyl)amino)isophthalic acid) [54].

The nanospecies [Zn(cpma)Cl_2_]_n_ (**21**) (cpma = 9,10-bis((4-carboxylatopyridinium-1-methylene)anthracene) was recently reported as a probe for live-cell imaging studies. The complex retained the viability of the human colorectal adenocarcinoma cell line (HCT-15) even at the highest concentration of 50 μM. Moreover, the fluorescent images showed that the CP was localized into the cell cytoplasm instead of the nucleus [55].

The metal–organic framework (MOF) [Zn_2_(1,4-bdc)_2_(4-bpdh)]_n_⋅3nDMF (**22**) (4-bpdh = 2,5-bis(4-pyridyl)-3,4-diaza-2,4-hexadiene) was obtained and proved to be efficient as a sensor for Fe(III) and Cd(II) ions with a high selectivity, excellent sensitivity, and short response time (<1 min). The DLs are estimated to be 0.2 and 0.5 μM, respectively. Moreover, this compound exhibits distinct solvent-dependent luminescent spectra with emission intensity significantly enhanced in dichloromethane [56].

The luminescent CPs [Zn_2_(3-bpat)_2_(1,3-bdc)_2_]_n_·nH_2_O (**23**), [Zn(3-bpat)(5-hip)]_n_·nH_2_O (**24**), and [Zn(3-bpat)(5-mip)]_n_ (**25**) (3-bpat = *N*,*N*’-bis(3-pyridylamide)-3,4-thiophene, 5-H_2_hip = 5-hydroxyisophthalic acid, 5-H_2_mip = 5-methylisophthalic acid) were prepared as optical sensors to detect target analytes. All species showed a good fluorescence quenching response to Fe(III) and MnO_4_^−^ with high selectivity and sensitivity, the DLs for both analytes being in the nanomolar range. Moreover, cation and anion competition experiments indicated that the luminescence quenching response of both analytes were almost unaffected by interfering ions [57].

[Zn_2_(tpom)(2,6-ndc)_2_]_n_·3.5nH_2_O (**26**) (tpom = tetrakis(4-pyridyloxymethylene)methane and H_2_ndc = 2,6-naphthalenedicarboxylic acid) was synthesized by a hydrothermal reaction, which has proved its utility as a selective sensor for the detection of Fe(III) and Cr(VI). Moreover, the sample resulting from Eu(III) incorporation into this species can differentiate between Fe(III) and Cr(VI) [58].

Highly selective sensing properties for Fe(III) and Cr_2_O_7_^2−^ ions were evidenced for {Zn(dhhpvb)(chdc)}_n_ (**27**), {[Zn(dhhpvb)_0.5_(oba)]·DMF·H_2_O}_n_ (**28**), and {[Zn(dhhpvb)_0.5_(sdb)]·H_2_O}_n_ (**29**) (dhhpvb = *E*,*E*-2,5-dihexyloxy-1,4-bis-(2-pyridin-vinyl)-benzene; H_2_chdc = 1,4-cyclohexanedicarboxylic acid, H_2_oba = 4,4′-oxybisbenzoic acid, H_2_sdb = 4,4′-sulfonyldibenzoic acid). Luminescence sensing behavior can be explained in terms of the competitive absorption of excitation wavelength energy between analytes and these species [59].

For luminescent pillar-layer {[Zn_2_(tda)_2_(azopy)_2_]·DMF}_n_ (**30**) (H_2_tda = thiophene-2,5-dicarboxylic acid, azopy = 4,4′-azobispyridine), the ability to act as a chemosensor for nitroaniline (NA) detection was highlighted. The luminescence response of (**30**) towards various aromatic compounds in DMF indicates average quenching Ksv values of 1.33 × 10^4^ (o-NA), 3.38 × 10^3^ (m-NA), and 2.75 × 10^4^ L mol^−1^ (p-NA), and DLs in the range 0.42–0.50 µmol L^−1^ [60].

Two zinc(II) crystalline CPs, [Zn_2_(cpif)_2_(bpe)(H_2_O)_2_]_n_⋅2nH_2_O (**31**) and [Zn(cpif)(bpp)]_n_ (**32**) (H_2_cpif = 5-(2-cyanophenoxy) isophthalic acid, bpe = 1,2-bis(4-pyridyl) ethane, and bpp = 1,3-bis(1-pyridyl) propane) (Figure 2a), synthesized under hydrothermal conditions exhibit good luminescent properties for rapidly detecting nitro-antibiotics, Cr_2_O_7_^2−^, and Fe(III) in water with high selectivity and sensitivity [61]. Sensing properties were also evidenced for {[Zn(bdmsb)(bpp)]·DMF}_n_ (**33**) and {[Zn(bdmsb)(bpe)]·DMF}_n_ (**34**) (H_2_bdmsb = 2,2′-[benzene-1,3-diylbis(methanediylsulfanediyl)]dibenzoic acid) with an excellent selectivity for Fe(III) and Cr_2_O_7_^2−^ ions in DMF. The quenching effects may contribute to the competitive adsorption of excitation wavelength energy between complexes and analytes [62].

Another series of CPs, [Zn(bptpa)(1,2-bdc)]_n_ (**35**), [Zn(bptpa)(1,3-bdc)]_n_·nH_2_O (**36**), and [Zn(bptpa)(1,4-bdc)]_n_·nH_2_O (**37**) (bptpa = *N*,*N*′-bis(pyridin-3-ylmethyl)-terephthalamide), prepared under hydrothermal conditions, shows multi-functional fluorescence responses for Fe(III), CrO_4_^2−^, Cr_2_O_7_^2−^, and pesticide 2,6-dichloro-4-nitroaniline (2,6-DC-4-NA), and good stability in a wide range of pH values [63].

The species with a 4,4-connected two-dimensional structure, [Zn_2_(mpnd)_2_(1,3-bdc)_2_]_n_ (**38**) and [Zn(mpnd)(hip)]_n_·3nH_2_O (**39**) (H_2_hip=5-hydroxyisophthalic acid, mpnd = *N*,*N*’-bis(4-methylenepyridin-4-yl)-1,4-naphthalene dicarboxamide), were synthesized under hydrothermal conditions. Both species exhibit multi-functional fluorescent responses towards Fe(III), MnO_4_^2−^, CrO_4_^2−^, Cr_2_O_7_^2−^, and 2,6-DC-4-NA with the DL in the range 10^−4^–10^−5^ M [64].

The MOF {Zn_2_(tpt)_2_(tad)_2_·H_2_O}_n_ (**40**) (tpt = 2,4,6-tri(pyridin-4-yl)-1,3,5-triazine, H_2_tda = 2,5-thiophene dicarboxylic acid) with a 3-fold interpenetrating 3D framework was successfully synthesized under hydrothermal conditions. Apart from its interesting structure, (**40**) displays excellent luminescence and stability in different pure organic solvents and acts as an ultrasensitive sensor for the detection of Fe(III) and picric acid (PA) [65].

An interesting Zn-MOF, [Zn(ebpba)(1,3,5-Hbtc)]_n_ (**41**) (ebpba = (E)-4,4′-(ethene-1,2-diyl)bis[(N-pyridin-3-yl)benzamide], H_3_btc = benzenetricarboxylic acid), can be used as a highly efficient fluorescence sensing material which provides a direct and low-cost method for the rapid detection of 3-nitrotyrosine (3-NT). Moreover, this species shows a high sensitivity with a KSV of 6.596 × 10^4^ M^−1^, a rapid response, excellent selectivity, high anti-interference ability, and good recyclability [66].

For the complex [Zn(tbta)(mbp)]_n_ (**42**) (H_2_tbta = tetrabromoterephthalic acid; mbp = 1,5-bis(2-methylbenzimidazol-1-yl)pentane), the solid-state ultraviolet emission was quenched by Fe(III) in aqueous solution with a detection limit of 0.85 μM [67].

Under solvothermal conditions, {[Zn_3_(bibz)_2.5_(ox)_3_(H_2_O)]·DMF·H_2_O}_n_ (**43**) {bibz = 1,4-bis(1-imidazolyl)benzene} was also prepared. The structural analysis revealed that (**43**) adopts a 2D framework, which is further expanded into a 3D supramolecular structure through CH⋯O hydrogen bonding. Luminescence properties reveal a highly sensitive and selective recognition for nitroaromatic compounds (nitrobenzene (NB), nitrotoluene (NT), nitrophenol (NP), and NA) and Fe(III) ions in aqueous solutions. Moreover, the luminescence quenching mechanisms were systematically revealed via a photoinduced electron transfer (PET) process, resonance energy transfer (RET) and fluorescence lifetime [68].

The complexes [Zn(opda)(pbib)]_n_ (**44**) (Figure 2b) and [Zn(pda)(pbib)(H_2_O)]_n_ (**45**) (H_2_opda = 1,2-phenylenediacetic acid, H_2_pda = 1,4-phenylene-diacetic acid, pbib = 1,4-bis(1-imidazolyl)benzene) were synthesized. The fluorescence property of both species allows the selective and sensitive detection of Cr_2_O_7_^2−^ and o-NP. The DLs are 2.992 × 10^−7^ and 4.372 × 10^−7^ M for Cr_2_O_7_^2−^, and 2.103 × 10^−7^ and 1.862 × 10^−7^ M for o-NP, respectively [69].

Compounds [Zn(dimb)(H_2_dobdc)]_n_·1.5nDMA·2.1nH_2_O (**46**) and [Zn(dimb)(H_2_dobpdc)]_n_·0.87nDMF·nH_2_O (**47**) (dimb = 1,4-di(1H-imidazol-4-yl)benzene, H_4_dobd = 2,5-dihydroxyterephthalic acid, H_4_dobpdc = 4,4′-dioxido-3,3′-biphenyldicarboxylic acid) adopt four-fold and three-fold interpenetrated networks and exhibit selective fluorescence quenching in NB even at low analyte concentrations. This behavior is associated with the presence of Lewis basic sites in the network [70].

Luminescent properties were also evidenced for the MOF [Zn_4_(OH)_2_(bdc)_3_(1,4-bip)_2_]_n_ (**48**) obtained through synthesis based on a mixed ligand strategy. The investigations indicated that it behaves as a sensor for a highly sensitive detection of TNP in DMF. The possible depletion in the photoluminescence intensity could be explained by a charge transfer process coupled with weak interaction between TNP and MOF [71].

The 3D MOF, [Zn_2_(4,4′-nba)_2_(1,4-bimb)_2_]_n_ (**49**), obtained from 3-nitro-4,4′-biphenyldicarboxylic acid (4,4′-H_2_nba) and 1,4-bis(imidazole-1-ylmethyl)benzene (1,4-bimb), proved to be a highly sensitive multi-responsive luminescent sensor for Fe(III), Cr_2_O_7_^2−^, and CrO_4_^2−^ in H_2_O and for NB in ethanol. Moreover, the compound can be recycled at least five times for sensing Fe(III) and Cr(VI) and can act as a fluorescent probe for Fe(III) under physiological pH conditions [72].

Ternary CPs, [Zn(pvbim)(1,2-bdc)]_n_ (**50**), [Zn_4_(pvbim)_2_(1,3-bdc)_3_(OH)_2_]_n_ (**51**), and [Zn(pvbim)(1,4-bdc)]_n_ (**52**) (pvbim = 2-(2-pyridin-4-yl-vinyl)-1*H*-benzimidazole), were recently proved as dual-functional luminescent probes. This ability was evidenced both for CPs and their paper-based sensors in the detection of levofloxacin (levo) and benzaldehyde (bzad) with high sensitivity and selectivity [73].

The luminescent MOF, [Zn(dptz)(1,4-bdc)(H_2_O)]_n_ (**53**) (dptz = 3,6-di(1*H*-pyrazol-4-yl)-1,2,4,5-tetrazine) (Figure 2c), exhibits excellent stability in the pH range 2–12. Furthermore, (**53**) can serve as a highly sensitive and efficient luminescent probe for detecting Fe(III) and Cr_2_O_7_^2−^ ions in aqueous systems. The values of KSV toward Fe(III) and Cr_2_O_7_^2−^ were estimated to be 1.61 × 10^3^ M^−1^ and 1.26 × 10^4^ M^−1^, respectively. The possible sensing mechanism consists of competitive energy absorption for both ions coupled with an interaction between the framework and the cation in the case of Fe(III) [74].

Another interesting derivative, 1-(triazol-1-yl)-2,4,6-benzene tricarboxylic acid (H_3_tzbt), was used to construct CPs {[Zn_2_(tzbt)(Htrz)]·3.5H_2_O}_n_ (trz = 1,2,4-triazol) (**54**) and {[Zn_2_(tzbt)(OH)(phen)]·4H_2_O}_n_ (**55**). Notably, they act as host for the encapsulation of Ln(III) ions and serve as an antenna to sensitize Tb(III) ions. By encapsulating different Ln(III) ions, both Eu_0.2_Tb_0.8_@(**54**) and Eu^3+^@(**55**) present white-light emission. In addition, (**55**) exhibits highly luminescent sensing properties in acetone [75].

The MOF [Zn(dpdc)(btb)_0.5_]_n_ (**56**) (H_2_dpdc = 3,3′-diphenyldicarboxylic acid and btb = 1,4-bis(1,2,4-triazol-1-yl)butane) adopts a 2D structure. The compound exhibits intense blue luminescence and acts as bi-functional chemosensor for Fe(III) and Al(III). The quenching mechanism of cations on the luminescence is assigned to a LMCT process [76].

The CP {[Zn_2_(btec)(trmb)_2_]·H_2_O}*_n_* (57) (H_4_btec = 1,2,4,5-benzenetetracarboxylic acid, trmb = 1,3-bis(1,2,4-triazol-4-ylmethyl)benzene) was synthesized and crystallographically fully characterized. It was demonstrated that the compound is a highly sensitive and selective luminescence sensor for the detection of Cr_2_O_7_^2−^, CrO_4_^2−^, and Fe(III) in aqueous solution with a DL of 3.05, 5.72, and 6.28 μM, respectively. This sensor is stable and can be recycled at least five times [77]. Another CP, [Zn_2_(1,4-bdc)_2_(dbpt)]_n_·2nMeOH (**58**) (dbpt = 2,7-di(4*H*-1,2,4-triazol-4-yl)benzo[lmn][3,8]phenanthroline-1,3,6,8(2*H*,7*H*)-tetraone), prepared using the same conditions, shows strong fluorescence and good stability in water. The chromate selectively quenches its fluorescence emission even in the presence of different anions [78].

Via the solvothermal method, [Zn_2_(Hcnoph)_2_(2,2′-bipy)_2_]_n_ (**59**) and [Zn(Hcnoph)(phen)(H_2_O)]_n_ (**60**) (H_3_cnoph = 3-[(1-carboxynaphthalen-2-yl)oxy]phthalic acid) (Figure 2d) were obtained as 1D-CPs. Moreover, both species exhibit excellent fluorescence performances, and a selective recognition response to Ni(II), MnO_4_^−^, Cr_2_O_7_^2−^, NB, and nitromethane through luminescence quenching effects in aqueous solution. The possible fluorescence recognition mechanism may be attributed to the synergistic effects of energy competitive absorption and weak interaction [79].

The 3D MOF [Zn_2_(Hddpb)(2,2′-bipy)]_n_ (**61**), synthesized by the V-pattern multi-carboxylic acid ligand 3,5-di(2′,5′-dicarboxylphenyl)benozoic acid (H_5_ddpb), presents high sensitivity for the detection of Fe(III) and Cr_2_O_7_^2−^ ions in aqueous solution [80].

The 3D MOF [Zn_2_(tcpbp)(4,4′-bipy)_2_]_n_ (**62**) (H_4_tcpbp = 2,2′,6,6′-tetra(4-carboxyphenyl)-4,4′-bipyridine), composed of four-fold interpenetrated diamond frameworks, was designed as a pH-sensitive luminescent system. The steric hindrance leaves the pyridyl groups uncoordinated, and thus accessible to the H^+^ ions. The compound suspension exhibits a reversible fluorescence transition in the pH range of 5.4−6.2 and can be used as a pH-triggered optical switch. The responsive mechanism consists of the disappearance of n → π* transitions and the appearance of IL transitions upon pyridyl protonation. Moreover, this MOF exhibits sensing ability for the detection of 3-nitropropionic acid, a major mycotoxin in moldy sugar cane [81].

Sensing properties were evidenced for [Zn_2_(cpota)(4,4′-bipy)(OH)]_n_·nH_2_O (**63**) (H_3_cpota = 2-(4-carboxyphenoxy)terephthalic acid) in the detection of hexavalent chromium anions in aqueous solution [82]. Strong luminescence was also observed for {Zn_2_Cl_2_(1,4-bdc)(4,4′-bipy)}_n_ (**64**), an ability developed for sensing Fe(III) and nitroaromatics such as NB, m-dinitrobenzene (m-DNB), p-NT, o-NT, p-NP, and o-NP in ethanol [83].

The 3D framework [Zn(Hntb)(phen)]_n_ (**65**) (H_3_ntb = 4,4′,4″-nitrilotribenzoic acid) was developed as dual-responsive fluorescent sensor toward NB and Fe(III) with a high selectivity, stability, and anti-interference ability [84].

The CP {[Zn(1,2,4-btc)(Hdpa)]·H_2_O}_n_ (**66**) (Hdpa = 4,4′-dipyridylamine) was synthesized under pH-controlled hydrothermal conditions. This complex exhibits excellent luminescence in both the solid state and in solution and can act as a multi-responsive luminescent sensor for Fe(III) and MnO_4_^−^ with high selectivity and sensitivity [85].

As a result, the proper selection of multifunctional organic ligands is very important for the preparation of Zn(II)-CBCPs, because it is crucial for obtaining structural diversity and good luminescent properties. Among numerous ligands, polycarboxylate, especially derived from rigid aromatic systems that exhibit both a large π-conjugated skeleton and the ability to coordinate as a bridge, are excellent candidates to construct such functional species. The properties can be further tuned by selecting a second ligand with the same characteristics, such as pyridine, imidazole, or triazole derivatives.

The proper selection of organic fluorophores (carboxylate and N-donor ligand) as rich aromatic moieties generally led to Zn-CBCPs with a broad range of emission energies. This aspect, together with pore dimensions, allow the specific recognition either of an inorganic or organic species, that can, in addition lead, to the possibility of developing sensors useful for a large variety of pollutant monitoring processes.

### 2.2. Zinc(II) Carboxylate Coordination Polymers with Catalytic Properties

Zinc is a metal with good catalytic performance based on its Lewis acid ability. Therefore, in the literature, many Zn(II) complexes have been reported with various types of ligands which are used for their catalytic properties. For instance, Zn(II)-CBCPs may catalyze the degradation of different organic dyes or influence the evolution of certain chemical reactions, as is presented in detail below.

Organic dyes and pharmaceuticals are identified frequently in water and wastewater, they resist biodegradation and present hazardous environmental effects. Among the reported technologies to remove these pollutants from waters, the use of photocatalytic degradation is a promising route to break down organic molecules into nontoxic species [132,133].

Consequently, {[Zn(1,4-bdc)(4,4′-bipy)]_n_} (**67**) and {[Zn(1,4-bdc)(Hyd)]_n_} (**68**) (Hyd = 8-hydroxyquinoline) have been reported [86] as efficient photocatalysts for the degradation of water pollutants, mainly of ibuprofen (IBP) whose presence is often identified in water and in wastewater treatment plant effluents. Over time, IBP degradation was investigated through different methods but the identification of the optimum method is still under study. The above-mentioned complexes were studied as heterogeneous photocatalysts in different conditions. Studies revealed that the photocatalytic degradation of IBP is more efficient in the presence of (**67**) under UV irradiation, whereas in the case of both complexes, the addition of H_2_O_2_ significantly decreased IBP levels.

Another Zn(II) CP, {[Zn_2_(1,4-bdc)_2_(1,3-bip)_2_]∙6H_2_O]}_n_ (**69**) (bip = 1,3-bis(2-methyl-imidazol-1-yl)propane), has been investigated [87] as a photocatalyst under UV light irradiation against methyl violet (MV) and rhodamine B (RhB). The results indicated that the MV photodegradation rate is slower than that of RhB and, in both cases, occurs only under UV influence, with the other reaction conditions being useless. Additionally, in the presence of (**48**), the MV degrades up to 81.5% over 40 min. In the photocatalytic process, the compound undergoes excitation to generate electron–hole pairs under visible light irradiation; the holes moves towards the Zn(II) center and the electron migrates to the ligand entity [71].

CPs [Zn(Hcpip)(2,2′-bipy)(H_2_O)]_n_ (**70**) and {Zn_3_(cpip)_2_(4,4′-bipy)_3_]_n_ (**71**) (H_3_cpip = 5-(4′-carboxylphenoxy)isophthalic acid) (Figure 3a) were reported [88] due to their photodecomposition properties on MV under UV irradiation. The performed studies showed that 64.82 and 78.18% of MV suffered photodegradation after 40 min of irradiation in the presence of complexes (**70**) and (**71**). In the absence of photocatalysts, degradation occurred for merely 18.66%. The higher efficiency of (**71**) as a photocatalyst is associated with the presence of 4,4′-bipy which led to the enhancement in the electron and hole transfer process.

Another complex, {[Zn(pmbd)(dpb)]∙dpb}_n_ (**72**) (H_2_pmbd = 3,3′-{[1,3-phenylene-bis(methylene)bis(oxy)}dibenzoic acid), has proven to be efficient as a photocatalyst for RhB, rhodamine 6G (Rh6G), and methyl red (MR) in aqueous solution under UV irradiation [89]. Hence, the degradation efficiency follows the order: RhB > Rh6G > MR, and according to developed measurements, the authors concluded that complex (**72**) is a suitable candidate for the photocatalytic decomposition of organic dyes.

A plethora of Zn(II) CPs were reported as photocatalysts for methylene blue (MB) degradation [90,91,134]. MB, used in the textile industry, is known for its carcinogenicity, non-biodegradability, and destructive effects on the environment [92] and, therefore, its degradation strategies have been studied intensively. For instance, (**42**) has proven its efficiency (93.9% under UV irradiation) for MB degradation [67]. Moreover, [Zn(pbta)_0.5_(bpa)]_n_∙2nH_2_O (**73**) (H_4_pbta = 5,5′-phenylenebis(methylene)-1,1′-3,3′-(benzene-tetracarboxylic) acid; bpa = 1,2-bis-(4-pyridyl)ethane) presented good MB photocatalytic degradation performances, with an efficiency of 91% (120 min after UV irradiation) [90]. Additionally, [Zn(1,2-bdc)(hmb)]_n_ (**74**) (hmb = 1,1′-hexane-1,6-diylbis(2-methyl-1H-benzimidazole)) and [Zn(pda)(hb)]_n_ (**75**) (H_2_pda = 1,4-phenylenediacetic acid; hb = 1,1′-hexane-1,6-diylbis(1*H*-benzimidazole)) were reported to catalyze MB degradation with an efficiency of 84.2% and 88.1%, respectively [91].

CP {[Zn_4_(glu)_3_(4,4′-bipy)_4_(H_2_O)_2_](NO_3_)_2_}_n_ (**76**) (H_2_glu = glutaric acid) was reported [92] to present the capacity to undergo photocatalysis for an anionic dye, methyl orange (MO). Degradation efficiency is associated with the particle size of the catalyst; therefore, it reached 11.71% for well-ground and 7.34% for large crystals. Additionally, authors [92] have evidenced that the adsorption capacity of MO molecules is promoted by electrostatic attraction between the anionic dye (MO) and the cationic framework of CP (**76**). This behavior also explains why complex (**76**) is unable to catalyze the degradation of MB, which is a cationic dye.

A micron-sized Zn(II) CP with the formula {[Zn(5-mip)(Bzp)]∙EtOH}_n_ (**77**) (Bzp = 1,3-bis(benzimidazole-1-yl)-2-propanol) was synthesized by an ultrasonic process and it has proven to be efficient as a heterogeneous catalyst for CO_2_ cycloaddition to epoxides at room temperature, the more so as CO_2_ is a very stable molecule [93].

An interesting and useful reaction in organic chemistry is Knoevenagel condensation, a method used to generate new carbon–carbon bonds which occurs with nitrogen-based catalysts [94]. The challenge for this reaction is to find the most efficient catalyst; hence, CP [Zn(paph)(NMeF)]_n_∙n(NMeF) (**78**) (H_2_paph = 5-{(pyren-4-ylmethyl)amino}isophthalic acid, NMeF = *N*-methylformamide) was under evaluation from this point of view in the Knoevenagel condensation of benzaldehyde and malononitrile in supercritical CO_2_ (scCO_2_) medium [95]. The evaluation of (**78**) as a catalyst in scCO_2_ was conducted using different co-solvents and the reaction yield increased from aprotic to protic co-solvents, the full conversion being recorded in the case of water.

Another exploited application of CPs is the possibility to act as catalysts in the electroreduction of CO_2_. This reaction consists of the conversion of CO_2_ to more reduced species, this being a promising approach to obtain fuels or other chemicals, or to reduce anthropogenic CO_2_ emissions [135]. In light of this information, complex [Zn_2_(daba)_4_(4,4′-bipy)]_n_ (**79**) (Hdaba = acid 4-diallylamino-benzoic) has proven a high efficiency and selectivity as an electrocatalyst for CO_2_ reduction to methanol, formaldehyde, and formic acid, which were identified by means of ^13^C NMR spectral data [96].

Having in view that CPs are used as electrocatalysts in the hydrogen evolution reaction (HER) due to their high surface area and active centers, some researchers [97,98] have synthesized the CP [Zn_2_(tzpi)(OH)(H_2_O)_2_]_n_∙2nH_2_O (**80**) (Figure 3b) using 5-(4-(tetrazol-5-yl)phenyl)isophthalic acid (H_3_tzpi), in order to investigate electrocatalytic performances. Consequently, they doped Co(II) ions into the CP framework to generate active sites so that the resulting product has a high specific surface area and presents excellent electrocatalytic properties for HER.

Mohammadikish and co-workers [136] synthesized a bi-metallic CP with a Schiff base ligand resulting from the condensation of a functionalized aldehyde and p-aminobenzoic acid, coded Zn-Mo-ICP (**81**) (ICP = infinite coordination polymer). The complex has proven its ability to adsorb MB (100%) and MO (52%), suggesting the tendency to more efficiently adsorb cationic dyes (MB). Moreover, complex (**81**) presents catalytic activity in the epoxidation of olefins with TBHP (*t*-butylhydroperoxide) as an oxidant, presumably also due to the existence of an N,O-donor Schiff base ligand. Studies have proved that the mentioned CP could be recycled four times without the loss of activity.

The exploitation of the Lewis acid character of Zn(II) afforded the development of several valuable materials with catalytic behavior. This ability was modulated through a large variety of carboxylate and N-donor ligands, the assembly of which generated pores able to specifically recognize organic species (dyes, drugs, etc.) as substrates. Coordination at Zn(II) centers led to substrate activation and, finally, their degradation.

### 2.3. Zinc(II) Carboxylate Coordination Polymers with Biological Properties

Nowadays, bacterial infections and multi-drug resistance have become a matter of global concern, with many complexes being screened in order to overcome this issue [137]. Among the evaluated complexes, Zn(II) containing ones are taken into consideration due to the biocidal effect of this ion [138].

Among CPs, [Zn_1.5_(CH_3_COO)_2_(4,4′-bipy)_2_]_n_(ClO_4_)_n_∙nH_2_O (**82**) has proven its biocidal activity against Gram-negative (*Escherichia coli*) and Gram-positive bacteria (*Staphylococcus epidermidis*) in both liquid and solid growth media compared with zinc sources, this behavior being related with a gradual and localized release of Zn(II) ions. Assessment of minimal inhibition concentrations (MIC) for compound (**82**) against *E.coli* and *S. epidermidis* led to values of 6.1 ppm and 4.6 ppm, respectively. The biocidal mechanism was associated with reactive oxygen species (ROS) generation and membrane disruption [99].

A series of CPs with indolecarboxylic acids were synthesized and subjected to biological tests [100]. Thus, complexes [Zn(I3aah)_2_]_n_ (**83**) (HI3aah = indole-3-acetic acid), [Zn(I3cah)_2_(H_2_O)]_n_ (**84**) (HI3cah = indole-3-carboxylic acid), [Zn(I3pah)_2_(H_2_O)]_n_ (**85**) (HI3pah = indole-3-propionic acid), [Zn(I2cah)_2_(H_2_O)_2_]_n_ (**86**) (HI2cah_2_ = indole-2-carboxylic acid) (Figure 4a), and [Zn(5-MeOI2cah)_2_(H_2_O)_2_]_n_ (**87**) (H5-MeOI2cah_2_ = 5-methoxyindole-2-carboxylic acid) were tested for antibacterial activity against Gram-positive (*Bacillus subtilis, Lysteria monocytogenes, Staphylococcus aureus*) and Gram-negative (*E. coli, Pseudomonas aeruginosa*) bacterial strains, and for antifungal activity against *Aspergillus niger* and *Candida albicans*. The results indicated that all complexes are more active than corresponding free ligands and present good activity against *B. subtilis.* Complex (**85**) is the most active against this strain (with an inhibition zone above 25 mm), followed by (**86**) (with an inhibition zone above 20 mm). Related to the activity against *L. monocytogenes*, complexes (**84**) and (**86**) exhibited inhibition zones above 20 and above 10 mm, respectively. Concerning the antifungal activity, the tests evidenced that from all tested complexes, only (**86**) presents activity against *A. niger*.

For [Zn(bfmta)(H_2_O)_2_]_n_ (**88**) (H_2_bfmta = 2,5-bis(furan-2-ylmethylcarbamoyl)terephthalic acid), biological tests were developed using Gram-positive (*B. subtilis, S. aureus*), Gram-negative (*E. coli, Salmonella typhi*) bacterial and fungal strains (*Penicillium expansum, Botrydepladia thiobromine, Nigrospora* sp., *Trichothesium* sp.) and were discussed in comparison with ciprofloxacin. The results evidenced that the antibacterial and antifungal activity of the complex is lower than of the standard drug but higher than of the free ligand (bfmta). The inhibition zones against bacterial strains are above 30 mm, whereas they are between 24 and 30 mm against fungal strains. Additionally, the antibacterial activity of (**88**) is higher than that of similar complexes with Mn(II), Co(II), or Ni(II) [101].

A different Zn(II)-CP, [Zn_4_(bdc)_4_(ppmh)_2_(H_2_O)]_n_ (**89**) (H_2_bdc = 1,4-benzene dicarboxylic acid; ppmh = *N-*pyridin-2-yl-*N’*-pyridin-4-ylmethylene-hydrazine), has proven its efficiency as an antibacterial agent against *E. coli* and *S. aureus*, being more active than the free ligands. As liver cancer is a malignant tumor with high incidence worldwide, finding new therapeutic agents has gained great importance. For example, complex (**89**) is also a potential antitumor agent against liver (HepG2) cell lines. The assay indicates that viability decrease in parabolic dependence with increasing concentration with an LD_50_ value of 42.2 ± 2.3 μg/mL. Ligands did not evidence any impact at levels up to 120 μg/mL and the positive control (cisplatin) had an LD_50_ value of 12.6 ± 2.8 μg/mL [102].

Complex [Zn(pab)(OH)(H_2_O)_2_]_n_ (**90**) has been reported [103] for its antibacterial activity against Gram-positive (*B. subtilis, S. aureus, Enterococcus faecalis*) and Gram-negative (*P. aeruginosa*, *Enterobacter cloacae*) strains, and activity was enhanced in comparison to free ligand (Hpab). The lowest MIC (12.5 μg/mL) was found for *E. faecalis, P. aeruginosa* and *E. cloacae*. Additionally, (**90**) presents antitumor activity against breast cancer (MCF-7), cervical cancer (HeLa), and cellosaurus (NCI-H446) cell lines.

For instance, another two complexes, {Zn_5_(pmbcd)_2_(μ_3_-OH)_2_(H_2_O)_4_(DMF)_2_]∙4DMF}_n_ (**91**) and {[Zn_2_(pmbcd)(bpa)_2_]∙2H_2_O∙2DMF}_n_ (**92**) (H_4_pmbcd = 9,9′-(1,4-phenylene bis(methylene))bis(9*H*-carbazole-3,6-dicarboxylate; DMF = *N*,*N*-dimethylformamide), were tested for the treatment of liver cancer by assessing the inhibitory effect against the viability of HepG2 cells [104]. The results demonstrated that complex (**91**) is more efficient than (**92**) and is a promising candidate for liver cancer treatment. Compound (**91**) inhibits prolyl hydroxylase-3 expression in the HepG2 liver cells significantly stronger than (**92**).

Moreover, complex {[Zn_2_(fum)_2_(Hdmpz)_4_]∙3H_2_O}_n_ (**93**) (H_2_fum = fumaric acid; Hdmpz = 3,5-dimethylpyrazole) (Figure 4b) was reported [105] due to its potential to act as an antitumor agent. The compound was evaluated against Dalton’s lymphoma malignant cancer. The mean IC_50_ value obtained by analyzing concentration–response curves of (**93**) and cisplatin was 19.21 and 0.45 μM, respectively. Additionally, complex [Zn(pna)_2_(H_2_O)]_n_ (**94**) (Hpna = 5-(pyrazol-1-yl) nicotinic acid) presents good anti-cancer activity on HeLa cells at a similar concentration to cisplatin, which is very valuable behavior having in view that, despite their efficiency, cisplatin and platinum analogs are cytotoxic and produce adverse reactions [106].

Five Zn(II) CPs with 5-azidoisophthalic acid (N_3_-H_2_ipa) and various N-donor co-ligands were synthesized by Mukherjee and co-workers [107] and tested for their cytotoxic effects on human colorectal carcinoma cell lines (HCT 116). Among these, [{Zn(H_2_O)_0.5_(N_3_-ipa)(phen)}]_n_ (**95**) has proven to be the most active with IC_30_, IC_50_ and IC_70_ values of 0.57, 25.56 and 50.55 μg/mL, respectively, which are much lower in comparison with other CPs.

Another CP, [Zn_3_(bib)(mtb)_2_]_n_ (**96**) (bib = 1,4-bis(benzimidazol-1-yl)-2-butene; H_3_mtb = 5-methoxybenzene-1,2,3-tricarboxylic acid), was tested for its anti-cancer ability on lymphoma cells [108]. The results indicated an excellent antitumor activity by increasing the cell apoptosis levels in combination with doxorubicin.

Anti-proliferation activity against human spinal tumor cells OPM-2 of complex [Zn(1,4-bdc)(bpybzimH_2_)]_n_(DMF)_0.5n_ (**97**) (bpybzimH_2_ = 6,6′-bis-(1H-benzoimidazol-2-yl)-2,2′-bipyridine) was reported as being superior to raw materials used for complex synthesis, this being a promising candidate for antitumor drugs. For instance, the IC_50_ value of (**97**) was 1.9 ± 0.05 μM, which is much lower than that of the positive control drug oxaliplatin (7.8 ± 0.2 μM) [109].

The complex [Zn_3_(ttha)(O)(OH)(H_2_O)_3_]_n_∙2nH_2_O (**98**) (H_3_ttha = 1,3,5-triazine-2,4,6-triamine hexaacetic acid) was evaluated as a treatment for gastrointestinal tumors and it was found that gastrointestinal function was promoted significantly after its application [110].

Treatment of glomerulus nephritis was efficient by the application of (**58**), this being evaluated by a reduction of accumulation of ROS in glomerular epithelial cells. The level of ROS decreased significantly after treatment with (**58**) in comparison to after treatment with the positive drug Nifedipine [78].

Zhou and co-workers [111] reported that [Zn(tptc)_0.5_(2,2′-bipy)(H_2_O)]_n_ (**99**) (H_4_tptc = *p*-terphenyl-2,2″,5″,5‴-tetracarboxylic acid) was able to reduce the weight and length of thrombus in animal models, the inhibitory effect being directly related with the dose.

Among various medicinal applications of zinc CPs, it is worth mentioning the therapeutic effect against coronary heart disease (CHD). Therefore, two complexes [Zn(H_2_cpn)_2_(H_2_O)_2_]_n_ (**100**) and [Zn(Hcpn)(2,2′-bipy)]_n_ (**101**) (H_3_cpn = 5-(3,4-dicarboxylphenoxy)nicotinic acid) were reported [112] for their potential in the treatment of CHD by decreasing inflammatory response in arterial endothelial cells and the platelet activation state. Studies indicated that (**100**) has a superior effect compared to (**101**).

Another complex [Zn(5-pro-ip)(bipr)]_n_∙2nH_2_O (5-pro-H_2_ip = 5-propoxyisophthalic acid; bipr = 1,3-bis(imidazolyl)propane) (**102**) was reported due to its effect on uterine fibroids [113].

Since Zn(II) ion is void of redox activity, the biologic behavior of compounds is also connected to its Lewis acid ability, enhanced by coordination to proper ligands. Furthermore, its stereochemical versatility allows the coordinative interaction with biomolecules (DNA, enzymes, etc.), followed by their splitting through hydrolytic processes.

### 2.4. Zinc(II) Carboxylate Coordination Polymers as Storage Materials

Besides interesting structural topologies, the design of CPs with a porous morphology has gained interest due to their ability to store different gaseous products, this being a manner of adsorbing inclusively greenhouse gases.

Although such complexes are compared to zeolites, the advantage over zeolites is the possibility to alter the architecture and functionalize the pores [139]. Storage properties are related with a high porosity and a high specific surface area. Additionally, this type of CP is able to trap organic molecules, including solvents [120], which is useful for separation purposes.

In such CPs, N-donor ligands combined with polycarboxylates as anionic linkers are used as a necessity to construct porous structures able to adsorb gases, which is illustrated in this section.

For instance, CP {[Zn(tptc)_0.5_(phen)]∙dioxane}_n_ (**103**) (Figure 5a) was synthesized with 25.3% porosity and presents gas storage properties [114]. It exhibits high CO_2_ selective absorption over N_2_ and CH_4_. Additionally, compared to zeolites, this complex exhibits a higher CH_4_ adsorption capacity.

Another CP, {[Zn_3_(pzdc)_3_(vim)_6_]∙vim∙2H_2_O}_n_ (**104**) (H_2_pzdc = pyrazine-2,3-dicarboxylic acid; vim = 1-vinylimidazole), was reported [115] due to the interesting coordination geometries of Zn(II) ions which, in the same polymeric chain, adopt distorted square pyramidal, distorted octahedral and distorted tetrahedral geometries. Additionally, despite the low surface area, the complex presents hydrogen adsorption properties.

Zhou and his group [116] reported the synthesis and characterization of complex [Zn_3_(Hbptc)_2_(e-urea)_2_]_n_∙2n(e-urea) (**105**) (H_4_bptc = biphenyl-3,3′,5,5′-tetracarboxylic acid; e-urea = ethyleneurea). It adopts a 3D porous framework with 1D nanotubes filled with e-urea molecules which are removed, and the resulting desolvated framework contains large pores (43% of the unit cell), this being suitable for gas adsorption. Tests evidenced that complex (**105**) presents moderate adsorption of H_2_ at 77 K and CO_2_ at 273 and 298 K, respectively.

Some CPs derived from 5-aminoisophthalic acid (H_2_aip) have been reported [117] due to their gas storage properties, in particular H_2_ and CO_2_. Hence, compounds {[Zn_2_(azpy)(aip)_2_]∙2DMF}_n_ (**106**) (azpy = 4,4′-azobipyridine; H_2_aip = 5-aminoisophthalic acid), {[Zn_2_(dipytz)(aip)_2_]∙1.15DMF∙0.85MeOH}_n_ (**107**) (dipytz = di-3,6-(4-pyridyl)-1,2,4,5-tetrazine), and {[Zn_2_(tpim)(aip)_2_]∙2.5DMF∙2H_2_O}_n_ (**108**) (tpim = 2,4,5-(tri(4-pyridyl)imidazole) present a honeycomb-like layer, with [Zn(aip)]_n_ pillared by azpy, dipytz and tpim, respectively, into a 2D porous framework, with 1D channels inside bilayers. The pore volumes decrease as follows: (**107**) > (**106**) > (**108**). Gas adsorption studies evidenced that (**108**) showed the greatest H_2_ and CO_2_ uptake abilities. Moreover, this adsorbs CO_2_ more efficiently than H_2_, with the taken gas amounts being 121.7 cm^3^∙g^−1^ and 55.7 cm^3^∙g^−1^, respectively.

CP [Zn_2_(N_3_-ipa)(4,4′-bipy)(DMF)_1.5_]_n_ (**109**) (N_3_-H_2_ipa = 5-azidoisophthalic acid) was tested for sorption properties of different gases (N_2_, O_2_, CO, CO_2_, C_2_H_2_) [118]. After drying at 120 °C in vacuum for 6 h, the compound resulted in being non-porous for N_2_, O_2_, and CO at 77 K. At higher temperatures, it was found that (**109**) was able to adsorb N_2_, O_2_, and CO (at 120 K), and CO_2_ and C_2_H_2_ (at 195 K). Furthermore, in situ photoirradiation enhanced its adsorption ability for O_2_.

Sezer and co-workers [119] synthesized and characterized a CP {[Zn(pda)(H_2_O)(bpa)Zn(pda)(bpa)]∙4H_2_O}_n_ (**110**) which presents high adsorption selectivity towards CO_2_ and the potential to be a good adsorbent material at 77 K. Additionally, it was found that the compound exhibits a higher adsorption selectivity for CO_2_/CH_4_ mixture against certain zeolites.

Another CP, {Zn(4-pyac)_2_}_n_ (**111**) (4-Hpyac = *trans*-3-(4-pyridyl)acrylic acid), has been reported [120] as the first diamantoid coordination framework able to trap *trans*-2-butene in square channels.

Besides the gas storage properties, CPs may present selective sorption properties of different organic compounds. Accordingly, Zaguzin and co-workers [121,122] reported complexes [Zn_2_(2-I-Pht)_2_bpa]_n_ (**112**) (Figure 5b), [Zn_2_(2-I-Pht)_2_dabco]_n_ (**113**), and [Zn_2_(1,4-bdc)_2_(dabco)]_n_ (**114**) (2-I-H_2_Pht = 2-iodoterephthalic acid, dabco = 1,8-diazabicyclooctane) for which the selectivity of sorption of different organic substrates from gas phases was tested. The results showed that (**112**) retained 1,2-dichloroethane from its mixture with benzene more readily in comparison with (**113**) and (**114**). Furthermore, (**112**) presented the best selectivity for benzene/cyclohexane mixtures, followed by (**113**) and (**114**).

The compound [Zn_2_dobdc]_n_ (**115**) (H_4_dobdc = 2,5-dioxido-1,4-benzenedicarboxylic acid) was developed as system that can adsorb octane [123], while [Zn_2_(H_2_O)(dobdc)]_n_·0.5nH_2_O (**116**) takes up a moderate amount of acetylene and performs a highly selective C_2_H_2_/CO_2_ separation [125].

Considering their mesoporous structures, these MOFs were tested for their potential to deliver IBP in phosphate-buffered saline solution by impregnating them with 30, 50 and 80 wt% IBP. The effect of IBP concentration on the release rate was evaluated and the maximum effective loading for each compound was estimated. The results evidenced that IBP loading was more effective in (**115**) in comparison with (**116**). More specifically, (**115**) could deliver a higher IBP concentration than (**116**) (50 vs. 30 wt%, respectively) at a faster rate. This study demonstrated that compound (**115**) is suitable for efficient drug delivery and dissolves in the process of drug release, this being an advantage for the drug delivery process [124].

A luminescent and nontoxic porous CP, [(Zn_8_(ade)_4_(bpdc)_6_O)]_n_·2nMe_2_NH_2_·8nDMF· 11nH_2_O (**117**) (H_2_bpdc = 4,4′-biphenyldicarboxylic acid, ade = adenine), was synthesized by a solvothermal process. The pores developed inside of the network provide both loading capacity (1.72 g g^−1^) and release ability (56% after two days) for diclofenac sodium [126].

These Zn(II)-based MOFs are studied as drug delivery systems due to their excellent biocompatibility, easy functionalization, high storage capacity, and excellent biodegradability [21]. In addition, considering that Zn-MOFs are developed based on a biocation, these are non-toxic species.

The modulation of porous cavities through carboxylate and ancillary ligands, the molar ratio and the reaction conditions allowed the development of a variety of materials able to selectively absorb and store certain organic and inorganic gaseous species, as well as carry drugs for a more efficient administration.

### 2.5. Zinc(II) Carboxylate Coordination Polymers for Miscellaneous Applications

In this section are gathered other applications of Zn(II)-CBCPs identified in the literature data besides the aforementioned ones.

For instance, such species can be used as anode material for lithium-ion batteries (LIBs). Taking into consideration that LIBs are used for various portable devices and electrical vehicles even in military purposes, this application gains a lot of importance.

Consequently, Xia and co-workers [127] synthesized compound [Zn(bcbpy)_0.5_(1,4-bdc)(H_2_O)]_n_ (**118**) (bcbpy = 1,1′-bis(3-carboxylatobenzyl)-4,4′-bipyridine) which was tested as an anode material for LIBs. The lithiation and delithiation processes were studied in the voltage range 0.01–3.0 V versus Li^+^/Li using cyclic voltammetry. Compound (**118**) can deliver a high reversible capacity of 386.2 mAh g^−1^ at 100 mA g^−1^ after 100 cycles. At a current density of 200 mA g^−1^, the capacity retention rate reached 93.1% after 1000 cycles.

Another group [128] reported compound [Zn(H_2_mpca)_2_(tfbdc)(H_2_O)]_n_ (**119**) (H_2_mpca = 3-methyl-1*H*-pyrazole-4-carboxylic acid; H_2_tfbdc = 2,3,5,6-tetrafluoroterephthalic acid) as the first 1D Zn-based CP that can serve as an electrode material for LIBs. Electrochemical performance as an electrode for (**119**) tested by galvanostatic cycling evidenced a great cycling stability and high reversible capacity of 300 mAhg^−1^ at 50 mAg^−1^ after 50 cycles. The authors also proposed a possible mechanism for electrochemical processes:$$\begin{array}{c} {}[\mathrm{Zn}(\mathrm{II}){\left(H_{2}\mathrm{mpca}\right)}_{2}(\mathrm{tfbdc})(H_{2}O)]+3{\mathrm{Li}}^{+}+3e^{-}\to [\mathrm{Zn}(\mathrm{II}){\mathrm{Li}}_{2}{\left(\mathrm{mpca}\right)}_{2}(\mathrm{tfbdc})]+\mathrm{LiOH}\\ \left[\mathrm{Zn}(\mathrm{II}){{\mathrm{Li}}_{2}(\mathrm{mpca})}_{2}(\mathrm{tfbdc})]+n{\mathrm{Li}}^{+}+ne^{-}↔[\mathrm{Zn}(0){\mathrm{Li}}_{n+2}{\left(\mathrm{mpca}\right)}_{2}(\mathrm{tfbdc})\right]\\ \mathrm{Zn}+{\mathrm{Li}}^{+}+e^{-}↔\mathrm{LiZn} \end{array}$$

Another application of Zn(II)-CBCPs is related to proton conduction properties, which present great importance for designing materials for catalysis and energy conversion systems [140].

For complexes {[Zn(pml)_0.5_(adth)]·5H_2_O}_n_ (**120**) and {[Zn(pml)_0.5_(adth)]·2H_2_O}_n_ (**121**) (H_4_pml = pyromellitic acid; adth = aldrithiol), proton conduction properties measured by alternating current impedance spectroscopy were reported. At 80 °C and 95% RH (RH = relative humidity), the investigated compounds present high conductivity values of 2.55 × 10^−7^ and 4.39 × 10^−7^ S cm^−1^. Additionally, time-dependent measurements evidenced the constant proton-conducting ability for both compounds up to 10 h. The authors of [129] stated that important roles in conductivity are played by the dimensionality of compounds and internal hydrogen bonding.

The compound PDMS-COO-Zn (PDMS = poly(dimethylsiloxane)) was reported for several applications. The tests evidenced that materials made from polymer present soft and viscoelastic properties at high temperatures, and due to rapid softening and hardening properties, it could be successfully used in medical fields (orthopaedic immobilization and external fixation systems). Another application of this species is in 3D printing, because when it is heated at 120 °C, it turns into a viscous liquid which quickly forms a rigid solid upon cooling. Additionally, by doping the compound with conductive materials, the resulting species are suitable for preparing conductive composites [141].

One-dimensional CP {[Zn(npdi)(1,3-bdc)]·H_2_O}_n_ (**122**) (npdi = 1,1′-(4-nitro-1,3-phenylene)bis(1*H*-benzo[d]imidazole) was used to synthesize multifunctional nanoparticles (NPs) (Ag, Au/Au_2_O_3_, Pd, Cr/Cr_2_O_3_/CrO_2_, Cu/Cu_2_O, Fe/FeO) at room temperature in the absence of a reducing agent [141]. The ability of this CP to be used in such a way arises from the ability of carboxylate at the inner surface of the cavities to anchor the metal precursors. The resulting NPs present various properties: Ag NPs present antibacterial properties; Ag, Au, and Cu NPs present ferromagnetic properties within the framework at room temperature. The same authors reported that compound (**122**) is able to sequester Cr(VI) toxic species into Cr NPs nontoxic ones [130].

Moreover, compound (**89**) was reported for Schottky diode barrier properties. Its conductivity is higher by one order in comparison with the free ligand (ppmh) (1.37 × 10^−6^ S cm^−1^ versus 6.2 × 10^−7^ S cm^−1^). The electrical conductivity is enhanced upon the irradiation of light. The specific detectivity of (**89**) is 11 times higher than that of the free ligand. All reported results suggest that this CP could be used for optoelectronic devices [102].

## 3. Conclusions

Relaying on the above-presented overview, it could be concluded that in recent years, a great number of Zn(II) carboxylate-based coordination polymers were developed by employing rigid carboxylate species alone or in combination with another rigid ligand, such as a heterocycle N-donor system. In these species, carboxylate ligands usually act as two, three or four-connectors, behavior related to the number of carboxylate groups, molar ratio, both substituents’ nature and ancillary ligands. The structure and morphology of these materials is difficult to control since it depends on various factors, such as the stereochemical versatility of Zn(II), nature and coordinative ability of the carboxylate and ancillary ligand, molar ratio, synthesis method, solvent nature, temperature and sometimes even pH. The 1D, 2D or 3D structures thus developed allowed luminescence manifestation, an ability used for the sensing and determination of a large variety of water pollutants, both inorganic (Cr(VI), Mn(VII), Fe(III), Cd(II), Cu(II), Ni(II), and Al(III) species) and organic (antibiotics and nitroaromatic derivatives). On the other hand, photocatalytic and biological applications are related to the Lewis acid ability of the Zn(II) ion. Considering this, photocatalysts active in dye (MV, RhB, MR, MO, and MB) or drug (IBP) removal from wastewater solutions were developed. Additionally, this ability led to the design of some Zn-CBCPs as inhibitors of several pathogenic microorganisms (Gram-positive and Gram-negative bacteria, and fungi), as well as tumors (hepatic, gastro-intestinal or lymphoma), insensitive to platinum-based drugs. The pores generated inside the network are suitable to include gaseous (H_2_, N_2_, O_2_, CO, CO_2_, CH_4_, and C_2_H_2_), liquid (1,2-dichloroethane, benzene, and cyclohexane) or solid (IBP and diclofenac sodium) species-generating materials valuable for storage, separation, or delivery. In conclusion, in this review we explored the vast and challenging domain of CPs and highlighted the main categories of properties for Zn-CBCPs, which are, in most cases, related with their structures.

## 4. Further Perspectives

The Zn-CBCPs described in this paper exhibit valuable properties and keep open the interest of researchers to obtain other similar species possibly with other applications that have been less addressed so far. As a result, the field can be developed both by using new polycarboxylate ligands with a rigid backbone and substituents with additional coordination ability, and by combining carboxylic derivatives with other polydentate ligands. The luminescent properties can be improved by including in the Zn-CBCPs network some known luminophores, such as lanthanide ions, for which several studies have been demonstrated so far, exhibiting their ability to modulate the optical properties of such derivatives. The domain of species with catalytic properties can be extended for other water pollutants and for other processes involving organic or inorganic species. Compounds with biological activity can also be tested for the inhibition of microbial biofilms or resistant tumors, aspects that have not been investigated so far in the field. In addition, the ability of some species to incorporate drugs could be explored for the development of another carrier, especially since the Zn(II) complexes are not toxic and have a low stability, thus easily releasing the active compounds. Finally, obtaining prosthetic materials or other opto-electrical properties can be further exploited in the future. The majority of the aforementioned species were synthesized mainly by the hydrothermal method, and it could be assumed that adopting other synthetic strategies may provide better control over the morphology. Therefore, even if in the past decade the domain of Zn-CBCPs was investigated in detail, the perspectives regarding their properties remain broad, with the possibility of improving existing ones or obtaining new candidates with new properties and applications. Hence, challenges still lie in the development of new species based on certain strategies which endow the CP with specific properties. The area of interest regarding the synthesis of new CPs will further gain importance as young scientists and researchers implement more efficient innovate methods with improved cost effectiveness.

## Data Availability

Not applicable.

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
