# Peer review of "Zinc(II) Carboxylate Coordination Polymers with Versatile Applications"

_molecules, 2023, doi:10.3390/molecules28031132_

Round 1

Reviewer 1 Report

Manuscript number: molecules-2156326

Title: Zinc(II) Carboxylate Coordination Polymers with Versatile Applications

Comments: The discussion and summary of zinc complexes in this paper have certain academic value in the field of new functional materials. Therefore, I recommend the publication of this manuscript in molecules. However, some revisions should be made.

1:  “n” in the molecular formula should be check and unified

2: “Further perspectives” should be reconsidered to give more constructive views.

3: After the list of examples, appropriate to add some comparison and discussion and other concluding discussion.

4: Relevant literature should be cited appropriately. Dalton Trans., 2022, 51, 11390–11396; Coordination Chemistry Reviews 475 (2023) 214892.

Author Response

Dear Reviewer,

Thank you very much for your valuable observations. Taking into account these aspects, we made the following modifications (highlighted in magenta) in the paper:

1: “n” in the molecular formula should be check and unified

“n” in the molecular formula was checked and unified both in text and Table 1

2: “Further perspectives” should be reconsidered to give more constructive views.

This aspect was reconsidered.

3: After the list of examples, appropriate to add some comparison and discussion and other concluding discussion.

Concluding discussions were added at each section.

4: Relevant literature should be cited appropriately. Dalton Trans., 2022, 51, 11390–11396; Coordination Chemistry Reviews 2023, 475, 214892.

The paper Dalton Trans., 2022, 51, 11390–11396 was cited together with corresponding discussion but Coordination Chemistry Reviews 2023, 475, 214892 (Anchoring polydentate N/O-ligands in metal phosphite/phosphate/phosphonate (MPO) for functional hybrid materials) has no connection with paper.

As result the references were renumbered and corrected in text and final list.

Reviewer 2 Report

1. All the figures only provide the structural feature, why not could be integrated with the functional properties, it will be interested for all the reader.

2. Herein, “Its stereochemical versatility allows also a structural arrangement in the complexes network that generates pores and thus, induces the ability of some materials to storage gaseous, liquid or solid species”, there are some documents could be cited, such as Micropor. Mesopor. Mat, 341(2022) 112098; J. Mater. Chem. A 4(2016) 11630-11634 and Inorg. Chem.,56(2017) 10215−10219 and Coord. Chem. Rev., 2021, 445, 214074

3. The review is organized into different parts, from the synthesis approaches to the different potential applications. A general discussion about the main advantages and current limitations of Zn-based MOFs in comparison with their others counterparts.

4. Same comment for the challenges associated to a fine characterization of the presence of the metals; a summary of the main recent progresses and remaining challenges of the usual characterization techniques should be given including recent publications.

5. When it deals with drug delivery, authors should better explain how Zn-based MOFs have their unique advantages in these aspects and provide some examples.

Author Response

Dear Reviewer,

Thank you very much for your valuable observations. Taking into account these aspects, we made the following modifications (highlighted in magenta) in the paper:

  1. All the figures only provide the structural feature, why not could be integrated with the functional properties, it will be interested for all the reader.

The functional properties were added in figures

  1. Herein, “Its stereochemical versatility allows also a structural arrangement in the complexes network that generates pores and thus, induces the ability of some materials to storage gaseous, liquid or solid species”, there are some documents could be cited, such as Micropor. Mesopor. Mat, 341(2022) 112098; J. Mater. Chem. A 4(2016) 11630-11634 and Inorg. Chem., 56 (2017) 10215−10219 and Coord. Chem. Rev., 2021, 445, 214074

All recommended paper were cited. As result the references were renumbered and corrected in text and final list.

  1. The review is organized into different parts, from the synthesis approaches to the different potential applications. A general discussion about the main advantages and current limitations of Zn-based MOFs in comparison with their others counterparts.

These was added in introduction part.

  1. Same comment for the challenges associated to a fine characterization of the presence of the metals; a summary of the main recent progresses and remaining challenges of the usual characterization techniques should be given including recent publications.

This part is very interested but was omitted considering that not cover the purpose of the paper.

  1. When it deals with drug delivery, authors should better explain how Zn-based MOFs have their unique advantages in these aspects and provide some examples.

The advantages of MOFs were added after the Zn(II)- carboxylate examples presented in paper.

Reviewer 3 Report

The review article submitted to molecules under the title "Zinc(II) Carboxylate Coordination Polymers with Versatile Applications" is a valued contribution and covers a broader range of literature for a specific topic. I do not have major concern against the article except a suggestion that authors may add a short section regarding the data collection. All keywords might also be included which they used during literature survey, and the mechanism that how they selected the literature for their article.  

Author Response

Dear Reviewer,

Thank you very much for your valuable observations. Taking into account these aspects, we made the following modifications (highlighted in magenta) in the paper:

I do not have major concern against the article except a suggestion that authors may add a short section regarding the data collection. All keywords might also be included which they used during literature survey, and the mechanism that how they selected the literature for their article.  

This section was added at the Introduction end.